# Pre-Clinical Investigation of Liquid Paclitaxel for Local Drug Delivery: A Pilot Study

**DOI:** 10.3390/ph13120434

**Published:** 2020-11-28

**Authors:** Claire V. Cawthon, Kathryn Cooper, Clifton Huett, Alyssa Lloret, Estefanny Villar-Matamoros, Lauren Stokes, Uwe Christians, Michele Schuler, Saami K. Yazdani

**Affiliations:** 1Department of Mechanical Engineering, University of South Alabama, Mobile, AL 36688, USA; clairecawthon@gmail.com (C.V.C.); kcg1101@jagmail.southalabama.edu (K.C.); cbh1523@jagmail.southalabama.edu (C.H.); 2Department of Engineering, Wake Forest University, Winston-Salem, NC 27101, USA; lloraa16@wfu.edu (A.L.); villare@wfu.edu (E.V.-M.); stokesl@wfu.edu (L.S.); 3iC42 Clinical Research and Development, University of Colorado, Aurora, CO 80045, USA; uwe.christians@cuanschutz.edu; 4Department of Comparative Medicine, University of South Alabama, Mobile, AL 36688, USA; mschuler@southalabama.edu

**Keywords:** peripheral arterial disease, liquid paclitaxel, local drug delivery, perfusion catheter, pre-clinical modeling

## Abstract

The purpose of this pilot study was to investigate the feasibility of a perfusion catheter to deliver liquid paclitaxel into arterial segments. A clinically relevant rabbit ilio-femoral injury model was utilized to determine the impact of liquid paclitaxel delivered locally into the vessel wall using a perfusion catheter at 1 h to 14 days. Treatment by two clinically available forms of liquid paclitaxel, a solvent-based (sb) versus an albumin-bound (nab), along with a control (uncoated balloons), were investigated. Pharmacokinetic results demonstrated an increase in the retention of the sb-paclitaxel versus the nab-paclitaxel at 1 h; however, no other differences were observed at days one, three, and seven. Histological findings at 14 days showed significantly less neointimal area in the sb-paclitaxel treated arteries as compared with the nab-paclitaxel and the uncoated balloon-treated arteries. Additionally, percent area stenosis was significantly less in the sb-paclitaxel group. These results support the concept of local liquid delivery of paclitaxel into the arterial segments.

## 1. Introduction

There are more than three million people affected by peripheral artery disease (PAD) every year in the U.S. Disparate from coronary artery disease, PAD is characterized by a substantial plaque burden and often presents with long and complex lesions [1,2]. Endovascular treatment of PAD focuses first on re-establishing blood flow by angioplasty and, more frequently, de-bulking by atherectomy. Angioplasty, described as a control injury, expands the lumen by outward stretching of the arterial wall. The vascular smooth muscle cells (VSMCs), residing in the vessel wall, respond to this injury by proliferating and migrating inward, re-narrowing the vessel lumen. This process is termed restenosis. To minimize the injury response and to inhibit VSMC proliferation, anti-proliferative drugs have been locally delivered using drug-eluting stents (DES) or drug-coated balloons (DCBs) [3,4].

Current endovascular treatments, including DES and DCBs, have limitations leading to inconsistent outcomes, patient readmission, and repeat revascularization. Stents tend to fracture due to the severe biomechanical environment of peripheral arteries [5,6,7,8]. Peripheral arteries undergo unique movements not observed in other parts of the body including compression, shortening, bending and twisting [9,10,11]. The use of a permanent device such as a metallic stent also limits re-interventional procedure in cases of restenosis.

To overcome the limitations of stents, DCBs were developed to locally deliver therapeutics to inhibit restenosis without the need of a permanent platform serving as a drug reservoir. DCBs rely on the transfer of the anti-proliferative drug, which is coated on the surface of the balloon, to the lumen of the arterial wall by simple contact of the balloon to the artery. DCBs are coated with a crystalline form of paclitaxel designed to adhere to the abluminal surface of the artery. Clinical studies have shown promising results with DCBs, but only in relatively short and simple lesions and mostly to treat disease in arteries above the knee [4,12].

In the periphery, paclitaxel has been the anti-proliferative drug of choice of DCBs due to its potency and binding affinity, making it an attractive drug for single use with long lasting effects [13,14]. Paclitaxel binds to tubulin, inhibiting depolymerization, which inhibits cell division and growth [15]. However, recent clinical publications have cast doubts on the safety profile of paclitaxel to treat PAD [16,17]. The INPACT-DEEP clinical study showed higher rates of amputation in the paclitaxel group than in controls, which was attributed to microembolization of particles related to means of drug attachment to the balloon [18]. Most notably the meta-analysis by Katsanos et al. showed paclitaxel-coated balloons and stents in femoral-popliteal arteries had an increased risk of all-cause mortality at three and five years [16]. Reviews of this analysis and the controversy surrounding it have encouraged further research into devices with alternative delivery systems [19]. Furthermore, Katsanos and his group published an additional meta-analysis study showing safety concerns for below-the-knee applications [17].

Current DCBs and DES use the dry (or powder) form of paclitaxel, which has a crystalline structure. Crystallinity of the paclitaxel drug increases its residency and local arterial pharmacokinetics and reduces solubility of the drug. This increase in residency is somewhat needed as the drug is positioned on the luminal surface and not within the medial wall. However, by placing the drug onto the luminal surface, there is a potential for the crystalline paclitaxel drug to become dislodged (mobilized), travelling to distal organs and tissue. Previously reported pre-clinical studies have demonstrated fibrinoid necrosis in downstream tissues of DCB-treated peripheral segments, along with higher paclitaxel levels in nontargeted tissue, suggesting increased emboli debris from crystalline paclitaxel [20,21].

In this pilot study, we investigated a new approach to deliver the therapeutic drug paclitaxel for peripheral applications. Specifically, we tested the use of a perfusion catheter to locally deliver the liquid form of paclitaxel directly into the vessel wall. Currently two available forms of liquid paclitaxel exist, a solvent-based (sb-) paclitaxel and an albumin-bound (nab-) paclitaxel, primarily used to treat cancer patients. The aim of this pilot study was to investigate the effectiveness of the solvent-based paclitaxel versus the albumin-bound paclitaxel form delivered locally by a perfusion catheter in a clinically relevant rabbit femoral-iliac injury model.

## 2. Results

### 2.1. Pharmacokinetic Analysis

In vivo studies were performed using a rabbit ilio-femoral injury model. The nab-paclitaxel and sb-paclitaxel were successfully delivered to the external iliac arteries using the perfusion catheter. An angiogram of the drug-filled perfusion catheter within the external iliac artery is shown in Figure 1. A total of 16 rabbits (32 artery segments) were used for pharmacokinetic analysis. Figure 2 summarizes the arterial concentration (ng/mg) results of the treated arteries. There was a significant increase in the retention of the sb-paclitaxel (*n* = 4) versus the nab-paclitaxel (*n* = 4) at 1 h (sb-paclitaxel: 4.106 ± 2.685 ng/mg vs nab-paclitaxel: 0.461 ± 0.270 ng/mg, *p* = 0.0004); however, no other differences were observed at days one, three, and seven. There was a significant drop in sb-paclitaxel between 1 h and one day (1 h: 4.106 ± 2.685 ng/mg vs one day: 0.108 ± 0.089 ng/mg, *p* < 0.001), but none between the other treatment time points.

### 2.2. Morphological and Histological Findings of Treated Arteries

Morphometric analysis demonstrated similar area measurements, including the external elastic lamina (EEL), internal elastic lamina (IEL), and lumen, for all treatment groups (Table 1). Medial area was significantly greater in the balloon-injured control group (no drug) as compared to the paclitaxel-treated arteries (uncoated balloon: 2.08 ± 0.47 mm^2^ vs sb-paclitaxel: 1.11 ± 0.26 mm^2^ vs nab-paclitaxel: 1.15 ± 0.28 mm^2^, *p* = 0.006). At 14 days, there was significantly less neointimal area in the sb-paclitaxel treated arteries as compared with nab-paclitaxel treated arteries and the balloon-injured control group (sb-paclitaxel: 0.26 ± 0.24 mm^2^ vs nab-paclitaxel: 1.15 ± 0.62 mm^2^ vs uncoated balloon: 1.42 ± 0.18 mm^2^, *p* = 0.007). Additionally, percent area stenosis was also significantly less in the sb-paclitaxel group (sb-paclitaxel: 8.37 ± 6.40% vs nab-paclitaxel: 22.60 ± 19.47% vs uncoated balloon: 33.08 ± 2.17%, *p* = 0.002). 

Histological analysis demonstrated no injury and no endothelial cell loss at 14 days (Figure 3). Inflammation was minimal for all groups and the presence of fibrin and platelets at the intimal layer was rarely observed (Table 1). Medial SMC loss, in the transmural direction, was observed only in the sb-paclitaxel treated group (sb-paclitaxel: 1.00 ± 0.82 vs nab-paclitaxel: 0.00 ± 0.00 vs uncoated balloon: 0.00 ± 0.00, *p* = 0.02). Medial SMC loss, in the circumferential direction, showed similar results (sb-paclitaxel: 0.75 ± 0.50 vs nab-paclitaxel: 0.00 ± 0.00 vs uncoated balloon: 0.00 ± 0.00, *p* = 0.007). No aneurysmal dilatation or thrombosis was observed in any treated artery.

## 3. Discussion

This study was designed to evaluate the efficacy of local liquid paclitaxel delivery to arterial segments in a clinically relevant rabbit ilio-femoral injury model. Two clinically available forms of paclitaxel, a solvent-based liquid paclitaxel and an albumin-bound liquid paclitaxel, were utilized to quantify the pharmacokinetics of paclitaxel retention within the arterial wall and histologically evaluate the treated arteries. Pharmacokinetic results indicated greater retention of the solvent-based liquid paclitaxel as compared with albumin-bound paclitaxel. Histologically, there was a significant decrease in neointimal hyperplasia and stenosis in arteries treated with the solvent-based liquid paclitaxel as compared to the albumin-bound liquid paclitaxel and the balloon-injured arteries. Additionally, drug-induced medial smooth muscle cell loss was more frequently observed in the solvent-based liquid paclitaxel treated arteries. These results indicate the effectiveness of a local liquid approach to deliver and retain liquid paclitaxel in arterial segments.

In this pilot study, the use of liquid forms of paclitaxel was investigated a potential therapy to inhibit restenosis. The selected intravenous (liquid) forms of paclitaxel have been utilized to treat patients with breast, ovarian, prostate and other forms of solid tumor cancers for the past four decades. In comparing the different forms of paclitaxel, the half-life of liquid (intravenous) paclitaxel is within hours [15,22], whereas the crystalline paclitaxel is weeks to months [14,23,24]. Thus, the potential for any lost (or mobilized) liquid paclitaxel to remain and accumulate within distal tissue or organs is minimal. Furthermore, the approach of liquid delivery offers an alternative to the current standards of DES and DCBs in which the anti-proliferative drug is deposited on the luminal surface. The described approach directly delivers paclitaxel into the medial layer, the residing region of proliferating smooth muscle cells.

Prior to testing, we performed bench-top studies to determine the optimal pressure range to deliver the liquid paclitaxel. The bench-top studies indicated that a pressure range of 0.1 to 0.4 atm was sufficient to drive the liquid paclitaxel sub-intimal, into the medial area (Figure 4). The built-in pressure sensor of the perfusion catheter enabled a quantifiable manner in which to ensure consistent and appropriate liquid paclitaxel delivery in our pilot study. The delivery pressure of the sb-paclitaxel treated arteries was similar to the nab-paclitaxel treated arteries (sb-paclitaxel: 0.32 ± 0.149 atm vs nab-paclitaxel: 0.37 ± 0.08 atm, *p* = 0.17).

Although all arteries were treated under similar conditions, the biological effect of paclitaxel was only observed in the sb-paclitaxel group. The difference in biological outcome, as determined by neointimal growth and vascular SMC loss, is most likely due to greater levels of paclitaxel retained in the arterial wall by the sb-paclitaxel group as compared with nab-paclitaxel. Axel et al. have previously demonstrated that paclitaxel, in a single dose-application of 20 min to 24 h, can inhibit vascular SMC growth up to 14 days. [13] Paclitaxel-induced SMC loss has been shown in arteries treated with paclitaxel-eluting stents and paclitaxel-coated balloons. Our results provide the first evidence that liquid paclitaxel delivered locally can induce biological effect.

Differences in the pharmacokinetic and biological outcomes of these two forms of liquid paclitaxel—solvent-based and albumin-bound forms—may be attributed to differences in solvents. Paclitaxel is extremely lipophilic and hydrophobic and therefore must be delivered in a mixture of solvents that allow it to be administered in the liquid form. The primary solvent in sb-paclitaxel is Cremophor EL (polyethoxylated castor oil), which surrounds the drug in a micelle and allows it to be transported in blood. The nab-paclitaxel does not require a solvent to solubilize the drug and is formulated with human serum albumin. Our results indicated greater acute retention of the sb-paclitaxel versus the nab-paclitaxel at 1 h (sb-paclitaxel: 4.106 ± 2.685 ng/mg vs nab-paclitaxel: 0.461 ± 0.270 ng/mg, *p* = 0.0004). The nearly 10-fold increase in paclitaxel measurement suggest the acute adhesion of the paclitaxel Cremophor EL to the extracellular matrix of the artery, consisting mostly of collagen, is greater as compared to the nanoparticle albumin carrier. A recent study demonstrated greater drug delivery of albumin-conjugated cancer drug when combined with a collagen-binding domain [25]. On the other hand, Cremophor EL has shown to increase paclitaxel retention, enhancing the ability of the paclitaxel to interact with tubulin and reducing cellular proliferation [26]. Further studies are warranted to elucidate differences in binding capacity of the Cremophor EL and the nano-particle albumin carrier to arterial wall collagen and elastin protein structures.

Although our studies were performed in clinically relevant models, we only restricted our studies to a healthy animal model and not one that is representative of patients with peripheral atherosclerotic disease. Additionally, the treated arteries lack major side branches, bifurcations, fibrosis, calcification, hemorrhage, and the need for debulking—all complexities that are often present in clinical settings. Lastly, appropriate controls and longer-time points are warranted to further demonstrate and characterize the impact of liquid paclitaxel onto arterial remodeling.

## 4. Materials and Methods

### 4.1. Perfusion Catheter and Liquid Paclitaxel Preparation

In order to deliver liquid paclitaxel locally to the selected ilio-femoral arterial segments, a multi-lumen balloon perfusion catheter was employed (Figure 5). The perfusion catheter (Occlusion Perfusion Catheter, Advanced Catheter Therapies, Chattanooga, TN, USA) is a universal delivery catheter that delivers therapeutic agents by creating a treatment chamber between two occlusion balloons, through which the liquid therapeutic is delivered. The delivery of the therapeutic agent is then mechanically driven, using pressure that is continuously monitored by a built-in sensor.

Two forms of liquid paclitaxel were delivered via the perfusion catheter—a nanoparticle albumin-bound (nab)-paclitaxel (Abraxane, Celgene, Summit, NJ, USA) and a solvent-based (sb)-paclitaxel (Paclitaxel Injection, Actavis Pharma, Parsippany, NJ, USA). The nab-paclitaxel is available in a solid form designed to be reconstituted with saline to make an injectable suspension. For this pilot study, the nab-paclitaxel particles were measured by weight and reconstituted with saline to a concentration of 6 mg/mL. The reconstituted nab-paclitaxel was then combined with saline and iohexol (Ominipaque, GE Healthcare, Wauwatosa, WI, USA) in a 1:2:2 ratio by volume (1-part nab-paclitaxel, 2-part saline, 2-part iohexol) to achieve a final paclitaxel concentration of 1.2 mg/mL. Similarly, the sb-paclitaxel purchased in liquid form (6 mg/mL) was combined with saline and iohexol in a 1:2:2 ratio by volume to achieve a final paclitaxel concentration of 1.2 mg/mL.

In our study, the perfusion catheter was used to deliver both forms of the liquid paclitaxel. The delivery of the liquid paclitaxel is accomplished by pressure differences created by an increase in treatment chamber pressure by the device, thereby driving the drug across tissue and into the medial layer. Because of this technique, it was not necessary for the therapeutic agent to be aspirated prior to removal of the device. Most importantly, this approach avoids the accumulation of drug on the luminal surface, as seen with drug coated balloons and drug eluting stents. Lastly, potential drug loss during tracking and positioning of the delivery device to the location of treatment is eliminated.

### 4.2. Rabbit Injury Model and Paclitaxel Delivery

This study protocol was approved by the University of South Alabama’s Institutional Animal Care and Use Committee (#568228) and conformed to the position of the American Heart Association on use of animals in research. A total of 16 rabbits (32 artery segments) were used for pharmacokinetic analysis, and a total of 6 rabbits (12 artery segments) were used for histological analysis. Two rabbits (4 artery segments) were allocated for each time point for both treatments. The experimental preparation of the animal model has been previously reported [27]. Briefly, following general anesthesia and endotracheal intubation, arterial access was obtained by cut-down approach and a vascular sheath was positioned into the carotid artery. Under fluoroscopic guidance, the animals underwent endothelial denudation (i.e., balloon injury) of both external iliac arteries using an angioplasty balloon catheter (3.25 × 6 mm). Subsequently, either sb- or nab-paclitaxel was delivered locally to the selected regions via the perfusion catheter (3 × 15 mm). In the balloon-injury control segments, no additional treatments was performed. Both external iliac arteries of each rabbit were treated with the same drug—either sb- or nab-paclitaxel. Liquid paclitaxel was delivered at a treatment chamber pressure range between 0.1 to 0.4 atm for 2 min. Antiplatelet therapy consisted of aspirin (40 mg/d), given orally 24 h before the procedure with continued dosing throughout the in-life phase of the study, and single-dose intra-arterial heparin (150 IU/kg) administered at the time of catheterization.

### 4.3. Pharmacokinetic Analysis

Following 1 h, 1 day, 3 days, and 7 days, animals were anesthetized and euthanized (intravenous FATAL-PLUS^®^, 85–150 mg/kg, single injection), and the treated artery segments were removed based on landmarks identified by angiography. The time points were selected to demonstrate the acute retention of liquid paclitaxel delivery. The explanted segments were then stored at −80 °C and shipped on dry ice to the bioanalytical laboratory (iC42 Clinical Research and Development, Aurora, CO, USA). As previously described, quantification of arterial paclitaxel levels was performed using a validated high-performance liquid chromatography (HPLC)-electrospray ionization- tandem mass spectrometry system (LC-MS/MS) [27,28,29].

### 4.4. Histological Analysis

Following the 14-day timepoint, animals were anesthetized and euthanized, and the treated artery segments were removed based on landmarks identified by angiography. The arteries were perfused by saline and formalin-fixed under physiological pressure prior to removal. The segments were stored in 10% formalin at room temperature and then processed to paraffin blocks, sectioned, and stained with hematoxylin and eosin (H&E) or Verhoeff’s elastin stain (VEG).

### 4.5. Histomorphometric Analysis

Histological sections were digitized, and measurements were performed using ImageJ software (NIH). Cross-sectional area measurements included the external elastic lamina (EEL), internal elastic lamina (IEL), and lumen area of each section. Using these measurements, the medial area (EEL-IEL), neointimal area (IEL-lumen), and percent area stenosis (100 × (IEL-Lumen)/(IEL)) were calculated as previously described [30,31,32].

Morphological analysis was performed by light microscopy using a grading criterion as previously published [30,31,32]. Parameters assessed included intimal healing as judged by injury, endothelial cell loss, intimal inflammation, and fibrin/platelet deposition. The medial wall was also assessed for drug-induced biological effect, specifically looking at smooth muscle cell loss, both in the transmural and circumferential directions. The presence of inflammation within the medial or adventitial regions was also evaluated. These parameters were semi-quantified using a scoring a system of 0 (none), 1 (minimal), 2 (mild), 3 (moderate), and 4 (severe) as previously described [30,32].

### 4.6. Statistical Analysis

All values are expressed as mean ± standard deviation (SD). Quantitative data were compared with analysis of variance (ANOVA), followed by Tukey’s test for multiple comparisons, using GrapPad Prism 8 (GraphPad Software, La Jolla, CA, USA). Non-parametric data were evaluated by Wilcoxon signed-rank test. A value of *p* < 0.05 was considered statistically significant.

## 5. Conclusions

Our overall results support the concept of local liquid delivery of paclitaxel into the arterial segments. In comparison of two clinically available forms of liquid paclitaxel, the solvent-based paclitaxel demonstrated greater arterial retention following treatment, accompanied with a decrease in neointimal growth. The histopathologic findings showed biological effect, as indicated by vascular smooth muscle cell loss, without evidence of thrombus, toxicity or inflammation. Together, these results indicate the effectiveness of a local liquid approach to deliver and retain liquid paclitaxel in arterial segments. Additional studies are warranted to further evaluate the safety and efficacy of the use of sb-paclitaxel in local liquid delivery devices—an approach that has the potential to impact millions of patients suffering with PAD by improving outcomes and quality of life.

## Figures and Tables

**Figure 1 pharmaceuticals-13-00434-f001:**
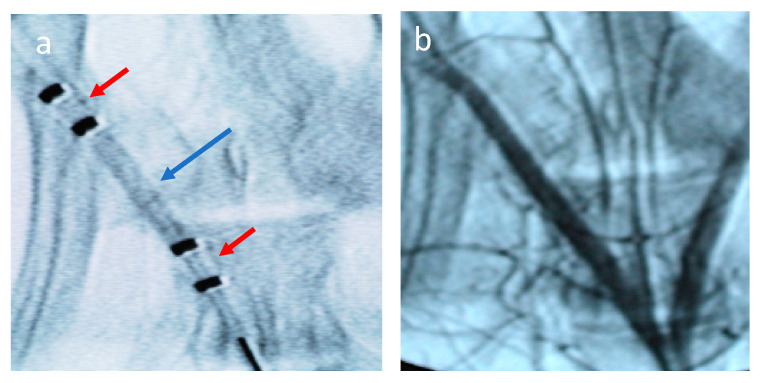
Angiogram of the perfusion catheter during delivery. (**a**) Liquid paclitaxel is shown filling the treatment chamber (blue arrow). The occlusion perfusion balloons are shown in the red arrows. (**b**) Angiogram following treatment of vessel by liquid paclitaxel.

**Figure 2 pharmaceuticals-13-00434-f002:**
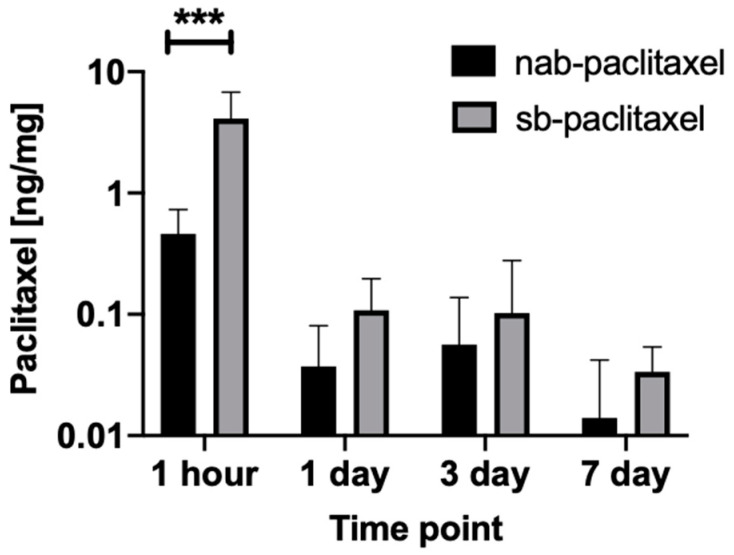
Mean tissue paclitaxel concentrations of the perfusion catheter treated arteries segments with the nab-paclitaxel and sb-paclitaxel liquid forms (bars represent standard deviation). A significant difference (*** *p* < 0.001) was observed at 1 h.

**Figure 3 pharmaceuticals-13-00434-f003:**
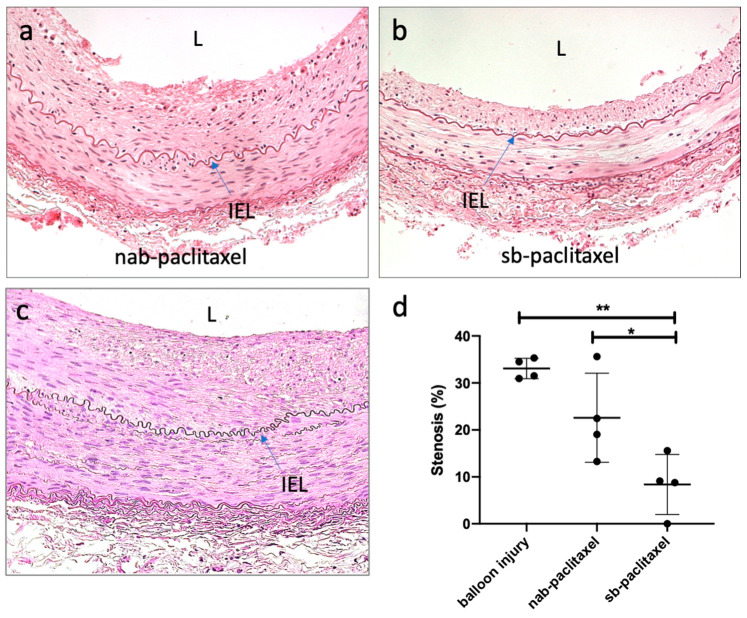
Histological assessment of liquid paclitaxel treated arteries. Representative hematoxylin and eosin (H&E) staining of (**a**) nab-paclitaxel, (**b**) sb-paclitaxel treated arteries, and (**c**) balloon-injured arteries. Blue arrows represent internal elastic laminae (IEL). Lumen is represented by L. (**d**) Significant differences (* *p* < 0.05, ** *p* < 0.01) in percent area stenosis were observed between the varying treatment groups (error bars represent standard deviation).

**Figure 4 pharmaceuticals-13-00434-f004:**
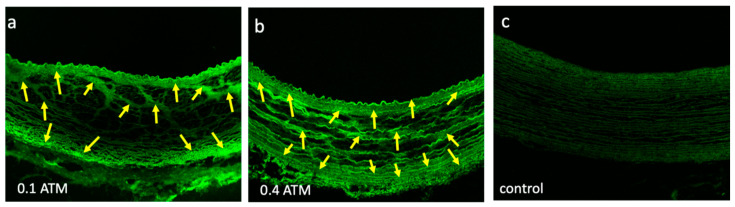
Liquid paclitaxel drug penetration. Confocal images of explanted pig arteries shows differences in Flutax-1 (fluorescently tagged paclitaxel, yellow arrows) penetration at (**a**) 0.1 atm and (**b**) 0.4 atm. (**c**) Control segment represents confocal image of a non-treated explanted pig artery.

**Figure 5 pharmaceuticals-13-00434-f005:**
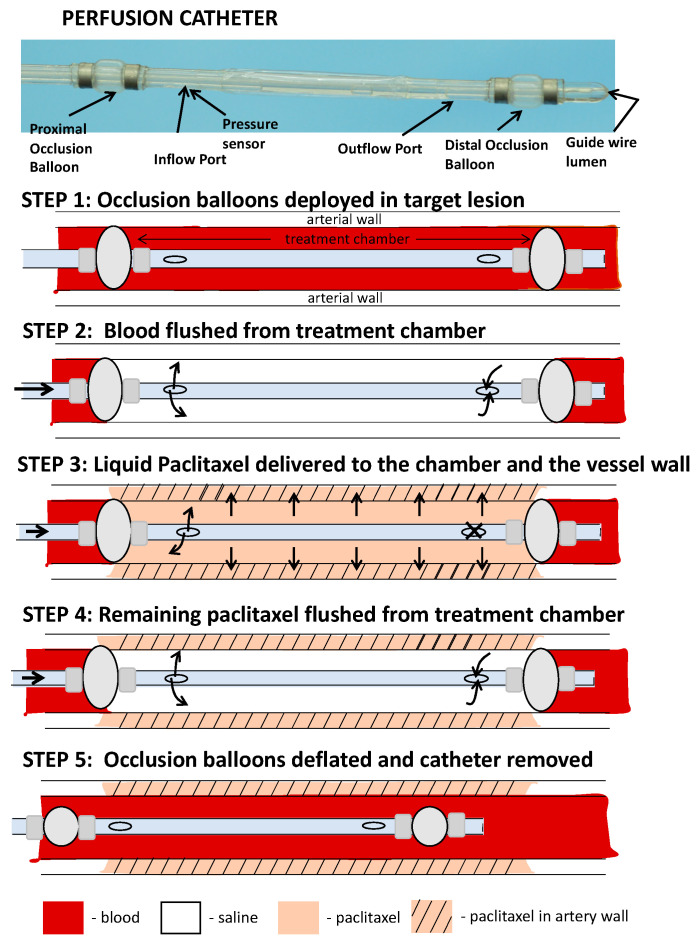
Schematic illustration of the perfusion catheter. (**step 1**) Two occluding balloons are deployed to momentarily stop circulating flow to a portion of a vessel. (**step 2**) Trapped blood within the perfusion chamber is then flushed with saline. (**step 3**) Liquid paclitaxel can be delivered through the inflow port. The outflow port is closed and drug delivered to the lesion with external pressure. The delivery pressure is measured by a fiber optic pressure sensor located at the inlet port. (**step 4**) Following the delivery of the drug, the remaining drug in the chamber is cleared through the outflow port to ensure no additional drug is introduced into the circulatory system. (**step 5**) The occlusion balloons are deflated and catheter moved to another location or removed from the circulation.

**Table 1 pharmaceuticals-13-00434-t001:** Summary of the morphometric and histological measurements in the rabbit iliac-femoral injury model.

Measurements	nab-Paclitaxel	sb-Paclitaxel	Uncoated-Balloons	*p* Value
*Morphometric Measurements*				
EEL, mm^2^	6.23 ± 1.42	4.92 ± 2.45	6.39 ± 0.68	0.43
IEL, mm^2^	5.10 ± 1.17	3.81 ± 2.31	4.31 ± 0.70	0.69
Lumen, mm^2^	3.93 ± 1.06	3.55 ± 2.19	2.89 ± 0.53	0.60
Media, mm^2^	1.15 ± 0.28 *	1.11 ± 0.26 *	2.08 ± 0.47	0.006
Neointimal area, mm^2^	1.15 ± 0.62	0.26 ± 0.24 *^,#^	1.42 ± 0.18	0.007
Percent area stenosis, %	22.60 ± 19.47	8.37 ± 6.40 *^,#^	33.08 ± 2.17	0.002
*Light Microscopy Analysis*				
Injury	0.00 ± 0.00	0.00 ± 0.00	0.00 ± 0.00	1.00
EC loss	0.00 ± 0.00	0.00 ± 0.00	0.00 ± 0.00	1.00
Inflammation (intimal)	0.50 ± 0.578	0.25 ± 0.50	0.00 ± 0.00	0.32
Inflammation (adv)	0.25 ± 0.50	0.50 ± 0.578	0.25 ± 0.50	0.75
Fibrin and Platelets	0.00 ± 0.00	0.25 ± 0.50	0.00 ± 0.00	0.41
SMC Loss (trans)	0.00 ± 0.00	1.00 ± 0.82 *^,#^	0.00 ± 0.00	0.02
SMC Loss (circum)	0.00 ± 0.00	0.75 ± 0.50 *^,#^	0.00 ± 0.00	0.007

Abbreviations: nab—albumin-bound, sb—solvent-based, EEL—external elastic lamina, IEL—internal elastic lamina, EC—endothelial cell, SMC—smooth muscle cell, adv—adventitia, trans—transmural, circum—circumferential, *—denotes significant difference as compared to uncoated balloon (control) group, ^#^—denotes significant difference as compared to nab-paclitaxel group.

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
