# Peer review of "Pre-Clinical Investigation of Liquid Paclitaxel for Local Drug Delivery: A Pilot Study"

_pharmaceuticals, 2020, doi:10.3390/ph13120434_

Round 1
Reviewer 1 Report
It gave me great pleasure to review the work conducted by Dr. Cawthon and his research group. In this paper, the authors describe the concept of local liquid delivery of paclitaxel into the arterial segments, post-acute arterial injury.
Major comments:
- This study had a limited sample size, making it challenging to draw any major conclusions. Therefore, I would recommend that the authors consider making this a pilot study and subsequently restructure the paper accordingly.
- The use of paclitaxel either through drug eluting balloons or stents in patients with peripheral arterial disease has been subject to significant debate. As the authors correctly mentioned in the introduction, this is a controversial area of research. For instance, Katsanos et al. demonstrated an increased risk of death following the application of paclitaxel‐coated balloons and stents in the femoropopliteal artery of the lower limbs. However, other studies (e.g. Schneider et al.) suggest otherwise. Therefore, I recommend that the authors further elaborate on the safety/benefits/advantages of their method of drug delivery over the current standards.
- Page 2 line 88: The authors induced an “acute injury” to the external iliac artery. However, PAD is inherently a chronic disease which involves atherosclerosis. Therefore, the “acute” arterial injury model that the authors are testing on does not necessarily mimic the chronic lesion/arterial injury observed in PAD. This makes it difficult to generalize/translate the study findings to PAD patients.
- Page 2 lines 88-89: Distal embolization is a major limitations of the current DCB/drug eluting stents. In this study, the authors did not investigate for traces of the drug in distal organs. This data needs to be demonstrated in order to justify the benefits of this current approach of drug delivery. I realize that that the authors comment on the potential accumulation of paclitaxel within distal tissue or organs (lines 133-136); however, this was not investigated within the study itself.
- Figure 1: is there an angio demonstrating the arterial injury? Was it an arterial dissection? Was the grade/degree of injury uniform among all patients? Is there any data which suggests that the type/degree of the injury might have an influence on the degree of PAC absorption, as this might be a confounding factor?
- Page 3 lines 112: This study lacks the presence of control groups. In my opinion, a positive control group with no treatment with PAC or balloon plasty post-injury is necessary here. This will help the reader comprehensively understand the benefits of PAC. A negative control treatment with plain balloon PTA would also be helpful while interpreting the data.
- The follow up period is fairly short (7 days),which makes it challenging to fully understand the potential side effects of the proposed method. Minor comments
- Figure 4: This figure is studied on pig’s artery which is not the same size/caliber as the animal model tested in this paper. Therefore, due to the size discrepancy, potential confounding might occur and influence the data. Why wasn’t the pressure tested on the same animal model?
- Lines 152-168: no comparisons were made between the dry form and the liquid form, preventing us from learning the benefits of one form over the other
Author Response
Submission Title: Pre-clinical investigation of liquid paclitaxel for local drug delivery: A pilot study
Answers to Reviewers’ Comments
Answers to Reviewer #1
We thank the effort of this Reviewer for all the valuable comments which have contributed towards improving our manuscript.
Comment 1: This study had a limited sample size, making it challenging to draw any major conclusions. Therefore, I would recommend that the authors consider making this a pilot study and subsequently restructure the paper accordingly.
Response: We agree with the reviewer and will present our data as a pilot study.
Comment 2: The use of paclitaxel either through drug eluting balloons or stents in patients with peripheral arterial disease has been subject to significant debate. As the authors correctly mentioned in the introduction, this is a controversial area of research. For instance, Katsanos et al. demonstrated an increased risk of death following the application of paclitaxel‐coated balloons and stents in the femoropopliteal artery of the lower limbs. However, other studies (e.g. Schneider et al.) suggest otherwise. Therefore, I recommend that the authors further elaborate on the safety/benefits/advantages of their method of drug delivery over the current standards.
Response: The following statement has been added to the Discussion in order to address this point:
“In this pilot study, the use of liquid forms of paclitaxel was investigated a potential therapy to inhibit restenosis. The selected intravenous (liquid) forms of paclitaxel have been utilized to treat patients with breast, ovarian, prostate and other forms of solid tumor cancers for the past four decades. In comparing the different forms of paclitaxel, the half-life of liquid (intravenous) paclitaxel is within hours [15, 22], whereas the crystalline paclitaxel is weeks to months [14, 23, 24]. Thus, the potential for any lost (or mobilized) liquid paclitaxel to remain and accumulate within distal tissue or organs is minimal. Furthermore, the approach of liquid delivery offers an alternative to the current standards of DES and DCBs in which the anti-proliferative drug is deposited on the luminal surface. The described approach directly delivers paclitaxel into the medial layer, the residing region of proliferating smooth muscle cells.
Comment 3: Page 2 line 88: The authors induced an “acute injury” to the external iliac artery. However, PAD is inherently a chronic disease which involves atherosclerosis. Therefore, the “acute” arterial injury model that the authors are testing on does not necessarily mimic the chronic lesion/arterial injury observed in PAD. This makes it difficult to generalize/translate the study findings to PAD patients.
Response: We agree that acute injury is not similar to the chronic disease process. An athero/disease model may be more representative of a PAD patient. However, in evaluating new catheter-based drug delivery technology, in particular for the first time, an injury model is preferred since it reduces the variations/factors associated with a disease model. This model is a preferred/recommended model set by the FDA in pre-clinical evaluation.
“Although our studies were performed in clinically relevant models, we only restricted our studies to a healthy animal model and not one that is representative of patients with peripheral atherosclerotic disease. Additionally, the treated arteries lack major side branches, bifurcations, fibrosis, calcification, hemorrhage, and the need for debulking—all complexities that are often present in clinical settings. Lastly, appropriate controls and longer-time points are warranted to further demonstrate and characterize the impact of liquid paclitaxel onto arterial remodeling.”
Comment 4: Page 2 lines 88-89: Distal embolization is a major limitations of the current DCB/drug eluting stents. In this study, the authors did not investigate for traces of the drug in distal organs. This data needs to be demonstrated in order to justify the benefits of this current approach of drug delivery. I realize that that the authors comment on the potential accumulation of paclitaxel within distal tissue or organs (lines 133-136); however, this was not investigated within the study itself.
Response: We agree that distal embolization was not evaluated. However, distal embolization due to drug coated balloons, and in some cases drug eluting stents, are associated with balloon coatings (excipients), and in DES polymeric coatings, dislodging and traveling downstream. In our delivery approach, liquid is delivered, so there are no solid materials, or dry coatings or polymers that can potentially cause distal embolization. We therefore did not evaluate distal organs and tissue for embolization.
Comment 5: Figure 1: is there an angio demonstrating the arterial injury? Was it an arterial dissection? Was the grade/degree of injury uniform among all patients? Is there any data which suggests that the type/degree of the injury might have an influence on the degree of PAC absorption, as this might be a confounding factor?
Response: Angio was performed during all procedures. No arterial dissections were observed. Additionally, we evaluated the treated sections histologically and showed no injury in any of the treatment groups (all IEL were intact). That is one benefit of the liquid approach in which low pressure is used to deliver the liquid therapy, whereas for a DCB, you must over inflate the balloon to approximately 20-25% greater than the diameter of the artery to achieve optimal drug delivery.
Comment 6: Page 3 lines 112: This study lacks the presence of control groups. In my opinion, a positive control group with no treatment with PAC or balloon plasty post-injury is necessary here. This will help the reader comprehensively understand the benefits of PAC. A negative control treatment with plain balloon PTA would also be helpful while interpreting the data.
Response: We agree with the reviewer. Our intention was to compare our results to a DCB-control. However, currently FDA-approved (clinically available) DCB are only available at 4mm x 40 mm in size (this is the smallest). This size is too large for the rabbit model and will result to sever dissection.
As suggested, we have included a treatment control injury group, with an uncoated balloon. We have modified Table 1 and Figure 3 accordingly.
Comment 7: The follow up period is fairly short (7 days),which makes it challenging to fully understand the potential side effects of the proposed method.
Response: We agree that the follow-up is short. However, for non-stent drug delivery devices, the acute drug retention is essential to characterize the device. For this pilot study, we therefore investigated the acute pharmacokinetics up to 7 days.
Minor Comment 1: Figure 4: This figure is studied on pig’s artery which is not the same size/caliber as the animal model tested in this paper. Therefore, due to the size discrepancy, potential confounding might occur and influence the data. Why wasn’t the pressure tested on the same animal model?
Response: We agree with the reviewer that a rabbit artery would have been ideal for comparison purposes. However, the pig carotid artery provides the longest arterial segment without branching, ideal for bench-top studies. In addition, pig carotid arteries are easily attainable from local slaughterhouses, saving resources.
Minor Comment 2: Lines 152-168: no comparisons were made between the dry form and the liquid form, preventing us from learning the benefits of one form over the other
Response: Our intention was to compare our results to a paclitaxel-coated DCB, which represents the dry form of paclitaxel. However, as stated previously, the currently FDA-approved (clinically available) DCB are only available at 4mm x 40 mm in size (this is the smallest). This size is too large for the rabbit model and will result to sever dissection. Our hopes are to be able to repeat these studies in a larger swine model, to be able to compare these studies head-to-head.
Reviewer 2 Report
Based on recent concerns regarding the safety of currently used formulations of paclitaxel for the local treatment of peripheral vascular disease, the authors used a perfusion catheter to compare the delivery of liquid paclitaxel to albumin-bound paclitaxel in a rabbit ilio-femoral arterial injury model. Overall, the data show that the liquid delivery of paclitaxel was superior in retention and in preventing restenosis.
Concerns:
The rationale for comparing liquid paclitaxel to albumin-bound paclitaxel is not clear given the stated rationale for safety concerns of the dry form of paclitaxel. Please provide a stronger rationale for comparing these two formulations with each other.
Table 1. Please provide data for comparison to a mock injured iliac-femoral artery and an injured iliac-femoral artery without paclitaxel.
It is recommended that summary data (Figures 2 and 3) be displayed as scatter plots.
What is the number of samples for the data in Table 1? Does the summary data represent SD or SEM? Given the SD in Table 1, was a power analysis performed to confirm that the conclusions (particularly for inflammation) are not subject to a type II error?
The authors conclude more medial smooth muscle cell loss with the solvent-based paclitaxel, but this data is not statistically significant.
It is not clear from the method description of how smooth muscle cell loss was quantified. Is this simply a decrease in the medial area? If so, how did the authors discriminate between changes in smooth muscle cell number and cell size?
The images in Figure 4 are difficult to interpret. Panels A and B are a black box, and panel C shows only the autofluorescence of the vessel wall. Each panel should demonstrate the autofluorescence of the medial layer. Arrows showing Flutax-1 would be useful.
Author Response
Submission Title: Pre-clinical investigation of liquid paclitaxel for local drug delivery: A pilot study
Answers to Reviewers’ Comments
Answers to Reviewer #2
We thank the effort of this Reviewer for all the valuable comments which have contributed towards improving our manuscript.
Comment 1: The rationale for comparing liquid paclitaxel to albumin-bound paclitaxel is not clear given the stated rationale for safety concerns of the dry form of paclitaxel. Please provide a stronger rationale for comparing these two formulations with each other.
Response: The forms of liquid paclitaxel, currently used for oncological purposes, were chosen as they are the two commercially available liquid paclitaxel available in the US. These two forms of paclitaxel, one solvent based and one albumin-bound based, provided two varying forms of liquid paclitaxel therapy to investigate.
Comment 2: Table 1. Please provide data for comparison to a mock injured iliac-femoral artery and an injured iliac-femoral artery without paclitaxel.
Response: We agree with the reviewer and have included this information in Table 1.
Comment 3: It is recommended that summary data (Figures 2 and 3) be displayed as scatter plots.
Response: We agree with the reviewer and have modified Figure 3 as a scatter plot. Figure 2 was maintained as a bar graph, since 3 of the 4 samples for the nab-paclitaxel at the 7-day time point was measured below the threshold of measurement using the mas-spec (so essentially no drug was measured for these samples).
Comment 4: What is the number of samples for the data in Table 1? Does the summary data represent SD or SEM? Given the SD in Table 1, was a power analysis performed to confirm that the conclusions (particularly for inflammation) are not subject to a type II error?
Response: We apologize if these were not made clear. There are 4 samples (N=4) for each data set for Table 1. The data represented in table 1 is mean +/- SD. With the addition of the control group, the data have been re-analyzed and p-values provided.
Comment 5: The authors conclude more medial smooth muscle cell loss with the solvent-based paclitaxel, but this data is not statistically significant.
Response: With the addition of the control group, the data have been re-analyzed and p-values provided (please see Table 1).
Comment 6: It is not clear from the method description of how smooth muscle cell loss was quantified. Is this simply a decrease in the medial area? If so, how did the authors discriminate between changes in smooth muscle cell number and cell size?
Response: Smooth muscle cell loss was performed by light microscopy analysis. This is a visual assessment of the medial layer of the treated segments, specifically looking for a lack of smooth muscle cells in this region. The scoring is based on a semi-quantitative scoring system, 0 = none identified; 1= minimal (<25%), 2 = mild (25 – 50%); 3 = moderate (50-75%); and 4 = marked (>75%). As mentioned, these have been described in previous publications.
Comment 7: The images in Figure 4 are difficult to interpret. Panels A and B are a black box, and panel C shows only the autofluorescence of the vessel wall. Each panel should demonstrate the autofluorescence of the medial layer. Arrows showing Flutax-1 would be useful.
Response: We agree with the reviewer. Panel C is indeed presented to show the autofluorescence of the vessel. Panels A and B show the fluorescence associated with Flutax-1. As suggested, we have included arrows showing flutax within panels A and B.
Round 2
Reviewer 1 Report
The authors have satisfactorily responded to all my questions and I congratulate them on their work.
Reviewer 2 Report
The authors have addressed my concerns.